# Exploring the Factors of Farmers' Rural–Urban Migration Decisions in Bangladesh

Abdullah Al-Maruf [1], A. K. M. Kanak Pervez [2], Pradip Kumar Sarker [3,*], Md Saifur Rahman [4] and Jorge Ruiz-Menjivar [5]

1 Department of Geography & Environmental Studies, University of Rajshahi, Rajshahi 6205, Bangladesh; ammaruf4@gmail.com
2 Department of Agronomy & Agricultural Extension, University of Rajshahi, Rajshahi 6205, Bangladesh; kp@ru.ac.bd
3 Forest and Nature Conservation Policy, Georg-August University, D-37077 Göttingen, Germany
4 Ministry of Public Administration, Bangladesh Secretariat, Dhaka 1000, Bangladesh; saifur69@yahoo.com
5 Department of Family, Youth and Community Sciences, University of Florida, Gainesville, FL 32611, USA; jorgerm@ufl.edu
* Correspondence: psarker@gwdg.de; Tel.: +49-551-39-33412

**Abstract:** In Bangladesh, rural–urban migration is widespread. Many earlier studies discussed the factors, patterns, causes, and consequences and the socio-economic and environmental impact of migration from the general perspective. However, rural–urban migration with a particular focus on particular communities or migrants' employment profiles, for instance, farmers, is poorly described. In contrast, many farmers move from rural to urban areas every year in Bangladesh. However, the factors that affect farmers' rural-to-urban migration are a primary concern to academia and key actors, as the country's economy mainly depends on agriculture and farming. This paper, therefore, aimed to identify the underlying factors of the rural–urban (R–U) migration of farmers in Bangladesh. Data for this study came from phone interviews conducted with 254 migrant farmers living in city districts in Bangladesh. We adopted a three-step approach to select and identify factors that impacted farmers' decision to move from rural to urban settings. First, we reviewed the extant literature and compiled more than 70 variables of interest relevant to farmers' migration. Second, 30 variables were selected for data collection after consultations with key informants (KIIs) and informal discussions (IDs) with farmers and local community leaders. Besides, the Q-methodology was used to assess the level of importance of the selected variables. Lastly, principal component analysis (PCA) was performed to extract salient dimensions of farmers' rural-to-urban migration, where 21 variables were detected that consistently exceeded a threshold value of 0.50 of communality for further analysis. Our findings show that six dimensions—i.e., individual, household, economic, attitudinal, spatial, and climate-induced extremes—significantly influence and contribute to rural urban migration decisions for farmers. Further, our results indicated that age, agricultural knowledge, household debt, seasonal famine/poverty (Monga), unemployment in rural areas, availability of anticipated job opportunities in urban areas, shortage of agricultural inputs, and river erosion significantly influenced farmers' decision to leave their farms in Bangladesh. Findings from this study may be used as inputs in predictive models and benchmark guidelines for assessing trends and patterns of rural-to-urban migration and for the formulation of policy and programs targeting domestic migration in Bangladesh for proper urban planning and further rural development.

**Keywords:** agricultural extension; key informants; informal discussions; principal component analysis (PCA); households; seasonal famine/poverty (Monga)

## 1. Introduction

Urbanization and industrialization have resulted in drastic economic development globally [1]. As a result, the income gap between the agricultural and non-agricultural

sectors has gradually expanded. Many countries have experienced considerable changes in the income and structure of rural households [2]. Consequently, the most significant aspect driven by the economic benefits is that many rural households go out to work due to the lack of agricultural laborers and the ageing of agricultural production in rural areas [3]. R–U migration can be defined as temporary or permanent migration from the countryside to urban cities, typically within the country's national boundaries [4]. Globally, about 740 million people are R–U migrants—four-times as many migrants who moved outside their countries [5,6]. The internal to international migration ratio in South Asia is comparatively higher than in other continents, about 10:1 [7]. Mainly, internal R–U migration is most prevalent in China and India, where the number of R–U migrants in these two countries exceeds the total number of migrants globally [8]. In Bangladesh, the increase in the urbanization rate reflects internal migration dynamics; from 1975 to 2009, this nation's urbanization rate rose by about 3%—one of the highest worldwide [9]. According to the World Bank data (see Table 1), from 1960 to 2018, the country's rural population doubled, while the urban population increased 23-fold due to R–U migration [10]. In 1960, the rural–urban population ratio was 18.47:1; however, by 2018, the rural–urban population ratio had notably changed to 1.73:1 [11]. Prior research has identified the rapid R–U migration as one of the vital determinants of the industrial development of Bangladesh [9,12,13]. Given that urbanization and industrialization are symbiotic processes [9], Bangladesh's urban population's accelerated growth has shifted from a mainly agricultural-based economy to one with a large and growing industrial sector. In 1961, the contributions of the agricultural and industrial sectors to Bangladesh's total gross domestic product (GPD) were 57.98% and 6.79%, respectively [14]. By 2018, the contributions of the sectors mentioned above to the national GDP have changed significantly, 13.07% and 28.53%, respectively [11,15,16].

Rural households' farmland abandonment has become a universal phenomenon of economic development in the world [17]. Previous research has examined labor migration and farmland abandonment in developing nations in Latin America, and Southeast Asia [18]. However, the literature related to labor migration in Bangladesh, particularly farmers from rural to urban areas, is limited [4].

Existing literature emphasized several factors that have contributed to the increase of R–U migration in Bangladesh; for instance, the textile industry in the country is the primary driver of within-country migration, where job opportunities in the ready-made clothing industry attract millions of rural people to urban areas, particularly women and the young, who seek financial independence [19,20]. Further, some studies focused on how coastal communities' vulnerability to climate change effects contributes to climate change displacement from rural to larger and urban cities [11]. According to Salam et al. [21], the Shock Index indicated that around 8.9% of the country's population lived in low-lying coastal, rural areas that were highly susceptible to frequent and intense weather events, such as typhoons, floods, and tsunamis. Estimates by the World Bank suggest that sea-level rise could result in up to 18% of coastal land being inundated and the displacement of 20–30 million coastal residents by 2100 [16,22]. Prior research has shown that R–U migration drivers include environmental, social, and economic challenges faced by farming communities in rural areas [23]. Other studies explored how crop seasonality, soil erosion problems, groundwater degradation, scarcity of water for drinking and irrigation, and climate-related financial risks to agricultural productivity are linked to rural poverty among farmers in Bangladesh [12,24]. R–U migration might be a livelihood strategy for smallholder farmers to secure stable job opportunities and higher income levels [25]. In Bangladesh, rural–urban migration is widespread. The factors, patterns, causes, and consequences and the socio-economic and environmental impact of migration from the general perspective are frequently discussed and cited [26,27]. However, research related to rural–urban migration with a particular focus on particular communities or migrants' employment profiles, for instance, farmers, remains limited. This paper aimed to explore and identify the underlying factors that significantly affect the rural–urban (R–U) migration of farmers in Bangladesh.

## 2. Theoretical Background

Neoclassical economies highlight the theoretical orientation of R–U migration. According to Ravenstein's (1885) [27,28] "laws of migration", most migration is from agricultural to industrial areas (Law 2), which combines individual rational choice theory with the broader structures of rural–urban and developmental inequalities, which is found in the much-vaunted push–pull framework. The push–pull theory by Lee (1966) [29] reveals migration to be driven by several push factors functioning from the region or country of origin (e.g., poverty, unemployment, landlessness, population growth, low social status, poor marriage prospects) and pull factors operating from the place or country of destination (e.g., better income, better education, and welfare systems, land to settle and farm, good environmental and living conditions, political freedom). In the context of migration, transitions, and development, very different from the individual-level rational choice decision-making of "neoclassical" migrants, is the broadly sweeping historical generalizations of Wilbur Zelinsky's "hypothesis of the mobility transition" (1971), as he pointed to early transitional society: mass rural–urban migration; emigration to attractive foreign destinations for settlement and colonization [30]. Todaro's model (1969) [31]— a model of R–U migration that seeks to interpret growing urbanization in developing countries—argues that migration is seen as an individual investment, increasing the possibility of acquiring a better job with a higher wage. Nevertheless, the interaction mechanism of labor migration from rural to urban, focusing on any particular community/group (e.g., farmers, teachers, fishers) in developing countries is not precise [17,32].

Migration studies from the perspective of developing countries have generally highlighted the economic aspects of migration [33]. Most of these investigations have emphasized the differentials and determinants of migration, mainly focusing on the causes and consequences of migration [34,35]. Regarding social and economic impacts, an individual's migration has demographic effects. The separation between members of households has contributed to low fertility rates in migrant families [36]. Some earlier studies focused on factors affecting the migration of household members from rural to urban and vice versa and the impact on land-use changes, urbanization, and food security from the general perspective [37,38]. However, limited studies have focused on adults, male and female [39], and tribal groups [40]. The earlier investigations widely analyzed rural laborers' decisions to migrate (participate in off-farm activities) and their destination (e.g., local employment, within the county, within the province) [37]. Though some studies have examined the time-use of off-farm jobs, they have mainly focused on proactivity (participating in both on- and off-farm work) [17]. They have usually paid more attention to the off-farm income of the whole household rather than the off-farm time-use of the individual [3]. Several studies focused on the correlations between rural–urban migration and land transfer. Some literature has distinguished the differences in factors that affect farmers' decisions in short-term and long-term off-farm employment, which are not clearly understood regarding the pattern of rural migration (e.g., rural–urban or urban–rural). The time-use of rural laborers' off-farm jobs points to their transition from the farm to the off-farm sector and shows the urbanization process at the micro-level [41]. Still, there are fewer empirical studies exploring the factors affecting particular groups' rural–urban migration, such as farmers [42]. Without a detailed categorization of the migration pattern of specific groups (e.g., farmers), critical information may be omitted when formulating labor and workforce policies [43].

Prior research has paid great attention to international migration patterns among Bangladeshis [8,44–47]. However, there is a lack of studies dealing with the dynamics of domestic migration, particularly with farming households. The current literature on internal migration has primarily focused on the effects of climate change and economic factors on coastal communities in Bangladesh [48–50]. The existing micro-level studies primarily investigated the characteristics of migrants at destination places, mainly Dhaka City, giving little attention to the causes of out-migration from rural areas [51]. Shams et al. (2014) studied the economic consequences of migration based on sample surveys conducted in Dhaka City [52]. Jahan (2012) found that out-migration is generally higher

in the villages characterized by land scarcity, unequal distribution of land, and a high proportion of agricultural laborers [53]. Haque and Islam (2012) pointed out that migrants often benefited more than nonemigrants because of their innovative and risk-taking nature [54]. The benefits included higher income, a gain in wealth, and greater access to public services and education. However, with a single study, most of the investigation was conducted on diverse people, such as daily laborers, slum dwellers, rickshaw pullers, and government and non-government professionals [55]. However, it is essential to give attention to rural–urban migration, particularly farmers, using micro-level studies based on sample surveys that have the advantage of identifying regional heterogeneity [19,56–59]. The current study sought to fill this gap by examining underlying multidimensional factors affecting farmers' decision to migrate from rural to urban cities in northwestern Bangladesh.

## 3. Materials and Methods

### 3.1. Sample and Data Collection

Data for this research were collected in subdistricts in Bangladesh: Gaibandha Sadar *Upazilla* (sub-district) in Gaibandha and Chilmari *Upazilla* in Kurigram (see Figure 1). The selection of these subdistricts was informed by discussions held with district office representatives from the Department of Agriculture Extension and prior studies indicating these areas have the highest percent of internal migrants [51,60,61]. To determine the sample size for data collection, we consulted household records at the Upazilla Agriculture Offices, which indicated that there were 2598 households from which at least one member has migrated from rural areas. Given time and financial costs, we randomly selected ten percent of these households for phone interviews—this approach met the necessary sample size of 243 individuals based on a 95% confidence level and a 6% margin of error [62]. The target sample size for interviews in each study site was 250 individuals, hence generating subsamples (representing between 8 and 10% of the population, respectively), allowing for relatively meaningful statistical analyses [63]. To (geographically) cover the different locations of the settled union area, field assistants—students and graduates of Rajshahi University—were asked to identify all mouzas across the union, allocate a similar target subsample to each, and approach households by moving from the center of the settlement (e.g., market square) towards the outer edges of the settlement along a major road/track and, depending on the size of the mouza, contact every third to fifth household (or the next household if no one was available at the targeted household). At least one migrant member in all 254 households agreed to participate in this study.

### 3.2. Variable Selection and Analysis

Identifying relevant and robust variables to understand the factors affecting farmers' rural–urban migration was a crucial step. To improve the suitability of the variables of rural–urban migration in the study area, the selected variables had to satisfy the following criteria: (i) the selected variables should be justified by previous studies on rural migration; (ii) the variables must be relevant to the scale of assessment (e.g., individual, households, and community); (iii) the variables must be measurable and easily interpretable; (iv) their measurement must be robust; (v) they should be used particularly in rural–urban migration farmers [62,63].

To ensure the criteria of variable selection, this study, therefore, employed a three-step approach to identify and select salient variables that affect farmers' decision to migrate internally from rural to urban cities in Bangladesh. First, we reviewed the extant literature related to R–U migration in Bangladesh from 2000 to 2020 via Google Scholar. We compiled a list of more than 70 variables that have been reported as influential factors for R–U migration. Next, we refined the list of variables in consultation with key informants (KII) and stakeholders from different Government and non-government organizations, such as the Department of Agriculture Extension, Union Parishad (lowest administrative unit of Bangladesh), Palli Karma-Sahayak Foundation (PKSF) (PKSF's overall goal is to create conditions for the people to move ahead not only in economic terms, but also in social terms

and in terms of increased capacity to deal with environmental problems), Gana Unnayan Kendra (GUK) (a group of dedicated social activists founded Gana Unnayan Kendra (GUK) in 1985 to reduce poverty by bringing positive, sustainable changes in the lives and livelihoods of disadvantaged communities), and Migrant Resource Centre Bangladesh, who has a wide range of experience on migration in Bangladesh, particularly rural–urban migration [32,64]. Specifically, we conducted five focus group discussions with Agricultural Extension officers from the research sites (i.e., Gaibandha and Kurigram districts).

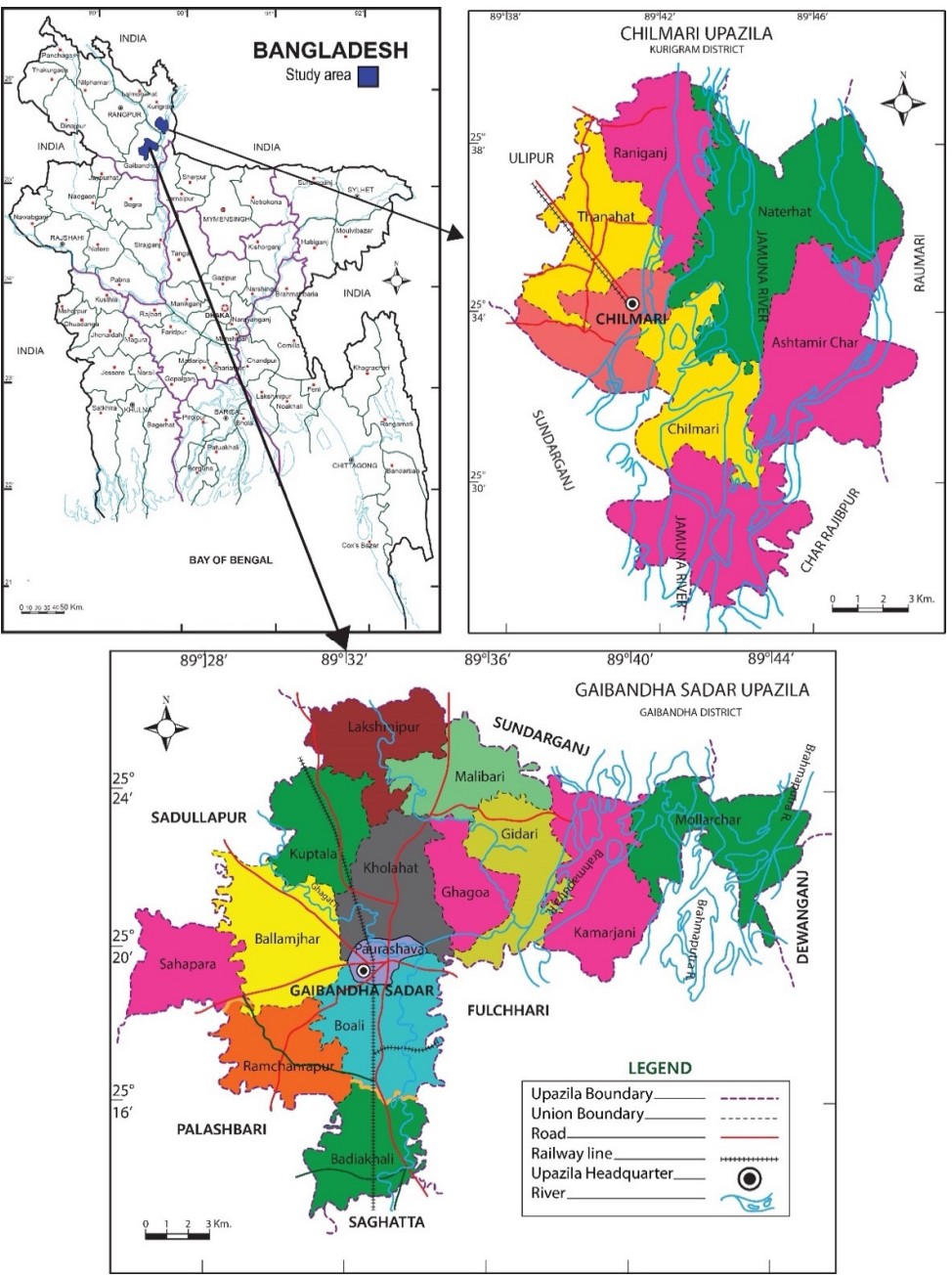

**Figure 1.** Research sites of the study. Source: own design.

We held formal and informal discussion meetings with local leaders and migrant farmers. The direct output of these focus groups and discussions was a list of 30 variables deemed influential in farmers' decision-making to migrate internally (Table 1). The variables and their measurement are shown in Table 1. Afterward, Q-methodology was used to assess the importance (dominance) level for variables influencing farmers' R-U migration

(Table 2). Lastly, principal component analysis (PCA) was performed to extract salient dimensions of farmers' rural-to-urban migration. The key variables of R–U migration as directed by the results of a median analysis were used in the principal component analysis with varimax rotation. The factor analysis technique was performed to extract and quantify the underlying factors of farmers' internal migration. Factor analysis identified patterns and revealed the underlying factors that accurately described the variation in the data [65]. Factor analysis was performed on each component to identify the variables with the highest variance in this study. The technique was used to investigate key traits from an array of overlapping relationships [66].

　　Several preconditions are needed to determine the suitability of PCA [67]. First, the sample size should include at least 200 participants. Second, the correlation matrix of the observed variables should display at least modest correlations to extract coherent factors [68]. In our research, the Kaiser–Meyer–Olkin (KMO) Measure of Sampling Adequacy ($\geq$0.60) and Bartlett's Test of Sphericity ($p < 0.001$) were used to test for the correlation among factors. We extracted only factors with an eigenvalue of 1.0 or more, in line with Kaiser's assumption. We reported factor loadings for each observed variable, indicating the correlations between the observed variables and the latent factors. A varimax rotation was performed to minimize the number of variables with high loadings on certain factors and to adjust for multiple scales in our measurement items. To understand the influence of variables on each component of R–U migration, we checked the communality for 21 variables that consistently exceeded a threshold value of 0.50 (Table 3). A reliability assessment was conducted to assess internal consistency to represent the overall decision of farmers' rural–urban migration. This helps check whether the components (e.g., individuals, household, economic, farmer's attitude towards farming, spatial, climate-induced extremes) had adequate precision. Cronbach's alpha coefficients were used to examine the reliability of each item. Cronbach's alpha coefficients can vary from 0 to 1, where 1 indicates perfect reliability and 0 indicates a very unreliable measure. In the early stages of research, a Cronbach's alpha coefficient approaching more than 0.60 is acceptable [69]. The study results found Cronbach's alpha coefficients of each item to be higher than 0.60. Thus, generally, these items indicate a relatively high level of internal consistency. The analysis of the data was conducted using SPSS 22.0.

**Table 1.** Description of selected variables and sources for assessing farmers' rural–urban migration.

| Selected Variables | Description of Variables | Scaling | Sources |
|---|---|---|---|
| Age | Years | 15+ | [70] |
| Education | Numbers of years of schooling | $\sum$(Statements) of years | [71,72] |
| Organizational participation | Access to government, non-government, and community-based organizations | $\sum$(Statements), each statement has a value of 0–3 (No = 0; Low = 1; Medium = 2; High = 3) | [62] |
| Cosmopolitanism | Frequent internal traveler compared to someone who has never visited any place | $\sum$(Statements), each statement has a value of 0–3 (No = 0; Low = 1; Medium = 2; High = 3) | [73,74] |
| Accessibility of mass media | Radio, television, local newspaper | $\sum$(Statements), each statement has a value of 0–4 | [75] |
| Agricultural knowledge | Access to knowledge of HYV, harvesting, seeding, pesticide, and land-use pattern | $\sum$(Statements), each statement has a value of 0–3 (No = 0; Low = 1; Medium = 2; High = 3) | [76] |
| Training received | Access to the various training programs | 1 for each training day | [71,77] |
| Family farm size | Total farm size | In hectares (ha) | [78] |
| Family income | Total monthly income | In thousand Bangladesh taka (BDT) | [79] |
| Family size | Number of family members | Number of family members | [80] |
| Family debt | Total monthly debt | In thousand Bangladesh taka (BDT) | [81] |
| Unemployment in rural areas | Status of unemployment in rural areas of farmers | Five-point Likert scale (0–4): very rare = 1; rare = 2: frequent = 3, and very frequent = 4 | [55] |

**Table 1.** *Cont.*

| Selected Variables | Description of Variables | Scaling | Sources |
|---|---|---|---|
| Income fluctuation | Stability of monthly income | Five-point Likert scale (0–4): very rare = 1; rare = 2: frequent = 3, and very frequent = 4 | [19,82] |
| Seasonal famine/poverty (Monga) | Effects of the pre-harvest period from September to November, plagued by seasonal hunger/poverty in the northwestern part of Bangladesh | Five-point Likert scale (0–4): very rare = 1; rare = 2: frequent = 3, and very frequent = 4 | [80,83] |
| Low price of agricultural products | Level of the price of agriculture products | Five-point Likert scale (0–4): very rare = 1; rare = 2: frequent = 3, and very frequent = 4 | [21,84] |
| Low agricultural wages in a rural area | Agricultural wage in rural areas of farmers | Five-point Likert scale (0–4): very rare = 1; rare = 2: frequent = 3, and very frequent = 4 | [85] |
| Low agricultural investments (e.g., seeds, fertilizer, cash, credit) from the Government | Access to agriculture investment by the government | Five-point Likert scale (0–4): very rare = 1; rare = 2: frequent = 3, and very frequent = 4 | [86] |
| Low/no turnover in farming | Supports (e.g., money, seeds, land) taken by individuals, GOs, and NGOs in a particular period | Five-point Likert scale (0–4): very rare = 1; rare = 2: frequent = 3, and very frequent = 4 | [87] |
| Perceived production risk | Low production due to seasonal variation and insects | Five-point Likert scale (0–4): very rare = 1; rare = 2: frequent = 3, and very frequent = 4 | [19,88] |
| Better communication infrastructure in urban areas | Better medium of transportation | Five-point Likert scale (0–4): very rare = 1; rare = 2: frequent = 3, and very frequent = 4 | [61] |
| Availability of anticipated jobs in the urban area | Job availability | Five-point Likert scale (0–4): very rare = 1; rare = 2: frequent = 3, and very frequent = 4 | [89] |
| Fascination with urban settings | Better livelihood facilities (e.g., schooling, health, and safe food) | Five-point Likert scale (0–4): very rare = 1; rare = 2: frequent = 3, and very frequent = 4 | [4] |
| Inadequate arable land | Status of cultivable land | Five-point Likert scale (0–4): very rare = 1; rare = 2: frequent = 3, and very frequent = 4 | [46] |
| Single-cropped area | The area only used for one crop | Five-point Likert scale (0–4): very rare = 1; rare = 2: frequent = 3, and very frequent = 4 | [90] |
| Low agricultural mechanization | Level of accessibility to agriculture machines | Five-point Likert scale (0–4): very rare = 1; rare = 2: frequent = 3, and very frequent = 4 | [25] |
| Low availability of agricultural inputs | Status of access to agriculture inputs (e.g., tractor, harvester, HYV seeds) | Five-point Likert scale (0–4): very rare = 1; rare = 2: frequent = 3, and very frequent = 4 | [91] |
| Seasonal flooding | Frequency of seasonal flooding | Five-point Likert scale (0–4): very rare = 1; rare = 2: frequent = 3, and very frequent = 4 | [60] |
| Drought | Intensity of drought | Five-point Likert scale (0–4): very rare = 1; rare = 2: frequent = 3, and very frequent = 4 | [26,92] |
| Abnormal rainfall and heatwaves | Period of rainfall and heatwave | Five-point Likert scale (0–4): very rare = 1; rare = 2: frequent = 3, and very frequent = 4 | [93,94] |
| River erosion | Frequency of river erosion adjacent to the settlement | Five-point Likert scale (0–4): very rare = 1; rare = 2: frequent = 3, and very frequent = 4 | [23,95] |

Compiled by the authors, 2021.

**Table 2.** Relatively important variables (based on the Q-methodology) influencing farmers' internal migration (*n* = 254).

| Degree of Importance | Variables |
| --- | --- |
| Most important | Agriculture knowledge, training received, seasonal flooding, river erosion, income fluctuation, organizational participation |
| Highly important | Inadequate arable land, seasonal famine/poverty (Monga), unemployment in rural areas, household income, low agricultural wage in rural areas, family farm size |
| Very important | Availability of anticipated jobs in urban areas, low availability of agricultural inputs, abnormal rainfall and heatwaves, household debt, age |
| Fairly important | Single-cropped area, low/no turnover in farming, perceived production risk, education |
| Less important | Cosmopolitanism, accessibility to mass media, family size, better communication infrastructure in urban areas, fascination with urban settings, low agricultural mechanization, drought, low agricultural investments (e.g., seeds, fertilizer, cash, credit) from the Government, low price of agricultural products |
| | G |

**Table 3.** Factor loadings and communality values of contributing variables under each factor (*n* = 254).

| Components | Contributing Variables of Components | Factor Loadings ($h^2$) | Communality |
| --- | --- | --- | --- |
| Individual | Age | 0.621 | 0.887 |
| | Organizational participation | 0.722 | 0.793 |
| | Agricultural knowledge | 0.853 | 0.885 |
| | Training received | 0.822 | 0.745 |
| Households | Family farm size | 0.773 | 0.743 |
| | Household income | 0.880 | 0.684 |
| | Household debt | 0.678 | 0.779 |
| Economic | Unemployment in the rural area | 0.889 | 0.780 |
| | Income fluctuation | 0.773 | 0.709 |
| | Low agricultural wages in rural areas | 0.674 | 0.774 |
| | Seasonal famine/poverty (Monga) | 0.765 | 0.782 |
| Farmers' attitude toward farming | Low/no turnover in farming | 0.785 | 0.674 |
| | Perceived production risk business | 0.590 | 0.689 |
| | Availability of anticipated jobs in an urban area | 0.664 | 0.779 |
| Spatial | Unavailable arable land | 0.870 | 0.773 |
| | Single-cropped area | 0.685 | 0.756 |
| | Low availability of agricultural inputs | 0.654 | 0.779 |
| Climate-induced extremes | Seasonal flooding | 0.653 | 0.664 |
| | Abnormal rainfall and heatwaves | 0.678 | 0.731 |
| | Drought | 0.596 | 0.699 |
| | River erosion | 0.775 | 0.880 |

## 4. Results and Discussion

### 4.1. Level of Importance (Dominance) for Variables Influencing Farmers' R-U Migration

Selected variables of farmers' R–U migration were assessed through median analysis; the median analysis of the sample of respondents assisted in identifying 30 key variables influencing farmers' R–U migration. Based on the Q-methodology, the study detected 20 "relatively important" variables (e.g., most, highly, very, fairly, less) influencing farmers' R-U migration (Table 2). The Q-methodology also elicited subjective viewpoints and identified shared patterns among the local farmers. The Q-methodology provides a solid basis for the systematic study of subjectivity [96]. Uniquely, the Q-methodology combines the strengths of both qualitative and quantitative methods. Typically, in a Q-methodological study, respondents are presented with a sample of statements about some topic (e.g., factors for rural–urban migration), called the Q-set. Respondents, called the P-set, are asked to rank-order the statements from their individual point of view, according to some preference, judgment, or feeling, mostly using a quasi-normal distribution. By Q-sorting, people

give their subjective meaning to the statements and, by doing so, reveal their subjective viewpoint. These individual rankings or viewpoints are then employed in the factor and median analyses [97].

As shown in Table 2, agriculture knowledge, training received, seasonal flooding, river erosion, income fluctuation, and organizational participation were the most important factors. Next, scarce arable land, poverty, unemployment in rural areas, household income, low agricultural wage, and family farm size were deemed highly significant factors of R–U migration. Moreover, our results indicated that the availability of anticipated jobs in the urban area, low availability of agricultural inputs, abnormal rainfall and heatwaves, household debt, and age played a significant role in migrating from rural to urban settings. Likewise, single-cropped areas, low/no turnover in farming, perceived production risk, and anticipated jobs in urban areas were important aspects that impacted farmers' decision to migrate internally. Importantly, our results suggested that better communication infrastructure in urban areas, fascination with urban settings, low agricultural mechanization, drought, education, cosmopolitanism, extensive media contact, low price of agricultural products, and low agricultural investment from the Government were less influential on farmers' decision to migrate to the country's cities.

### 4.2. Underlying Factors Affecting Farmers' Internal Migration

This section presents and summarizes the results of factor analysis. Factor loadings and communalities are reported in Table 3, and eigenvalues and the percent contribution of factors to the total variance are reported in Table 4.

**Table 4.** Eigenvalue and percent contribution of factors to the total variance (*n* = 254).

| Components | Eigenvalue (λ) | % of Variance |
|---|---|---|
| Individual | 2.603 | 15.44 |
| Household | 2.531 | 15.02 |
| Economic | 2.120 | 12.58 |
| Farmers' attitude toward farming | 1.873 | 11.11 |
| Spatial | 1.632 | 9.68 |
| Climate-induced extremes | 1.541 | 9.14 |

#### 4.2.1. Factor I: Farmers' Individual Characteristics

Based on PCA analysis, the first component of R–U migration can be explained through four variables: age, organizational participation, agriculture knowledge, and training received. The communality for extracted variables was 0.687, 0.793, 885, and 0.745 with factor loadings of 0.621, 0.722, 0.853, and 0.822, respectively.

Older farmers tend to have a relatively lower probability of engaging in rural-to-urban migration. Al-Maruf (2017) found almost similar findings that farmers aged 20–50 are much more interested in rural-to-urban migration due to greater physical capability [60]. Farmers with higher levels of agriculture knowledge create job opportunities in urban areas, which plays a crucial role in R–U migration. Organizational participation may contribute to farmers' migration—typically, growers receive information from such organizations about occupations and jobs requiring agricultural skills in urban areas, creating migration channels. As such, farmers with high organizational participation represent a relatively higher percentage of R–U migrants [91]. Our results also indicated that farmers who have received training from governmental and not-for-profit organizations are more likely to migrate from rural to urban areas [71]. Trained farmers are exposed to a wide range of ideas and knowledge, such as seeding, pesticides, irrigation, and crop selection [59,62]. Training on seeding, planting, fertilizer use, harvesting methods, and land-use decisions are the type of training that enhance farmers' urban working competitiveness. Table 4 shows that individual characteristics explained a higher percentage of the total variance (15.44), indicating that this particular dimension is vital for explaining R–U migration in Bangladesh.

### 4.2.2. Factor II: Household's Characteristics

The second contributing dimension explaining rural–urban migration is household characteristics. This dimension consisted of three variables: family farm size, household income, and household debt with communality values of 0.743, 0.684, and 0.779 and factor loadings of 0.773, 0.880, and 0.678, respectively. In rural areas of Bangladesh, more than 70% of producers are smallholder farmers [98]. They tend to migrate to bear their family's basic needs and expenditures. Therefore, farmers with small farms show a more migratory attitude than large-sized farm holders. Higher household income levels may ensure their overall livelihood expenditures and essential better services such as school, college, hospitals, other public authorities, and household members, which led to R–U migration. Household debt is another push factor for farmers for R–U migration. Hence, they have to cover their households' expenditures through debt. To reduce debt pressure, farmers migrate from rural to urban areas. Other household members contribute to family expenses through the return of money from the migrated members of the family [19]. The household characteristics dimension explained 15.02% of the variance in R–U migration.

### 4.2.3. Factor III: Economic Characteristics

The third component of farmers' internal migration is economic characteristics. Economic characteristics include four variables, namely unemployment in rural areas, income fluctuation, low agriculture wages in rural areas, and poverty. Employment opportunities are very limited in rural areas in Bangladesh. Hence, most marginal and middle-sized farm holders are interested in migrating from rural to urban areas through short-term and seasonal migration. However, overall income fluctuates due to limited employment opportunities and natural disasters. Farmers migrate internally to deal with this economic challenge [89]. Low agriculture wage is another critical factor for internal migration. Based on the informal discussion, the study found that although some farmers find limited employment opportunities in rural areas, wages are meager. Thus, this crucially influences migration from rural to urban areas, as Mallick [99] found. The current study also revealed that many marginal farmers migrate from rural to urban due to poverty [100]. The economic characteristics contributed 12.58% to the total variability of the data.

### 4.2.4. Factor IV: Farmers' Perception/Attitude towards Farming

The fourth contributing component of farmers' internal migration was attitudes towards farming, which included the following variables: low turnover, perceived production risk, and anticipated jobs in urban areas with communality and loadings of 0.674, 0.689, and 0.779 and 0.785, and 0.590 and 0.664, respectively. Rural farmers' overall income negatively affects local farming, as most farmers are considered to have a low turnover rate (Mallick et al., 2015). In addition, current challenges in agricultural production, including the increased price of farming inputs, poor soil quality, crop disease, low sale prices, poor revenue for farmers, and environmental shocks, contribute to the notion of agriculture as a highly risky activity [27,63,101]. As a result, farmers may migrate to urban areas and engage in other non-agricultural sectors (e.g., tertiary economic activities). Many farmers in rural Bangladesh are involved in subsistence farming, where most food crops produced are grown to provide for the basic food needs of the household with a small surplus for commercialization [90]. Thus, farmers, especially those not engaged in commercial agriculture, move to urban areas to seek job opportunities with relatively higher salaries than in rural areas. Farmers' perception/attitude toward farming shows an overall variance of 11.11%, which moderately influences rural–urban migration among farmers.

### 4.2.5. Factor V: Spatial Characteristics

Lake and Fenner [86] pointed out that land quality, spatial crop pattern, and water availability influence R–U migration in rural Bangladesh. In our study, spatial characteristics comprise three variables: unavailability of arable land, single-cropped area, and low availability of agriculture inputs—these factors have communality values and factor

loadings of 0.773, 0.756, 0.779 and 0.870, and 0.685 and 0.654, respectively. This dimension explained about 9.68% of the total variance. In Bangladesh, two-thirds of the population lives in villages, where approximately 41.54% of the economically active population is engaged in agriculture, contributing 12.64% to the country's total GDP [102]. In the absence of stringent land-use policies in rural areas, farming land has drastically decreased and has been transferred to commercial, residential, industrial, and other unplanned uses [103]. Moreover, the land-use and availability issues are exacerbated by the low availability of agricultural inputs coupled with higher price tags and the lack of crop diversification [27].

4.2.6. Factor VI: Climate-Induced Extremes

Climate-induced extreme events, for example, floods, cyclones, and droughts are predicted to increase in Bangladesh [6]. The effects of climate shock negatively impact ecosystems, exacerbate the local water crisis and land degradation, and threaten the livelihoods of millions of rural residents and farming communities in Bangladesh [49]. Smallholder farmers are among the most vulnerable groups to climatic risks as they face chronic poverty and food insecurity [102]. The sixth influential dimension contribution to R–U migration is climate-induced disasters. This dimension consisted of four variables: seasonal flooding, abnormal rainfall and heatwaves, drought, and river erosion. The communality values and factor loadings for seasonal flooding, irregular rainfall and heatwaves, drought, and river erosion were 0.664, 0.731, 0.699, and 0.880 and 0.653, 0.678, 0.596, and 0.775, respectively. This dimension explained only 9.14% of the total variability in the data.

## 5. Conclusions and Recommendations

The main objective of this paper was to identify the underlying factors in the rural–urban (R–U) migration of farmers in Bangladesh. For instance, the factors, economic, demographic, social, and environmental, that affect this R–U migration are multiple and complex in nature [104]. However, our findings indicated that factors drive farmer's R–U migration in six main dimensions: individual, households, economic, farmer's attitude towards farming, spatial, and climate-induced disaster. Further, our results indicated that agriculture knowledge, training received, seasonal flooding, river erosion, income fluctuation, and organizational participation are the most influential factors that affect R–U migration, as cited by Kumar et al. [105]. In addition, age, agricultural knowledge, household debt, seasonal famine/poverty (Monga), unemployment in rural areas, availability of job opportunities in urban areas, shortage of agricultural inputs, and river erosion significantly influenced farmers' decision to leave their farms in Bangladesh.

Since rural–urban migration mainly originates from a lack of rural economic development, creating jobs and other opportunities for earning income in rural regions can address the current problem [106]. There should thus be three different kinds of investment in the rural areas, as also noted by Walter [104]. First of all, to increase agricultural production, the Government should continue to promote the modernization of the agricultural sector. More efforts to adapt to environmental risks and diversify agricultural output will also increase agricultural productivity. The State needs to research and develop more flood-tolerant, drought-resistant, heat-resistant, and salinity-resistant crops and train farmers to grow them, providing reliable seeds at a moderate cost. Secondly, Bangladesh should develop rural industries and infrastructure and upgrade health and learning in rural areas, so that farm families believe do not need to migrate to cities to avail themselves of these services. Subsidies in rural areas should be encouraged. Guaranteeing balanced national development requires economic growth's regionalization (i.e., decentralization). Political parties have successfully campaigned on this issue for 30 years and on the related issue of moving the capital out of Dhaka to stop concentrating development there, but no action has been taken. Establishing industrial estates, education facilities, and private investment in semi-urban centers may accomplish this. Lastly, an integrated R–U migration strategy is necessary, and these policies should be designed to work on the underlying causes of

rural–urban migration. Such a strategy must be proactive in addressing current problems and also forestalling future ones.

The Bangladesh Government published a draft of the country's national urban policy in 2011 for the coming years, entitled "National Urban Sector Policy" draft [104]. This document recognizes the current spatial imbalance in the pattern of urbanization in the country and also the drastic pace of urban population growth in Dhaka and other big cities. It also shows both positive and negative results of R–U migration. R–U migration needs to be guided appropriately to build balanced urbanization to avoid mass population aggregation in one or few cities [107].

The Department of Agricultural Extension (DAE) should take necessary actions to provide practical training to the rural farmers to increase their agricultural knowledge and increase the overall yield of crops. Furthermore, the Government should create an insurance scheme against agricultural risks, such as floods, disasters, and crop losses due to pest infestations. Concerning the prevention and management of natural disasters, short-term strategies include constructing additional drainage systems that disperse water surpluses and minimize floods. In addition, improving flood protection is an essential issue in large metropolitan centers. To halt the present degree of disorganization in flood prevention and reduction, the Government should establish a hierarchical chain of command for a Department of Disaster Management to govern.

**Author Contributions:** Conceptualization, A.A.-M. and A.K.M.K.P.; methodology, A.A.-M. and A.K.M.K.P.; formal analysis, A.A.-M.; field investigation and interviews, A.K.M.K.P.; resources, A.A.-M. and A.K.M.K.P.; writing—original draft preparation, A.A.-M. and P.K.S.; writing—review and editing, A.A.-M., A.K.M.K.P., P.K.S., M.S.R. and J.R.-M.; visualization, A.A.-M. All authors have read and agreed to the published version of the manuscript.

**Funding:** This research received no external funding.

**Institutional Review Board Statement:** Not applicable.

**Informed Consent Statement:** Not applicable.

**Data Availability Statement:** Supporting data of the findings of this study are available from the second author upon reasonable request.

**Acknowledgments:** We are very grateful to the GO's and NGOs' executive officials and the local farmers who provided insightful information regarding rural–urban migration. The authors are, of course, responsible for errors of facts and interpretation. We acknowledge the Open Access Publication Funds by the Georg-August University, Göttingen, Germany.

**Conflicts of Interest:** The authors declare no conflict of interest.

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
