# Peer review of "Exploring the Factors of Farmers’ Rural–Urban Migration Decisions in Bangladesh"

_agriculture, doi:10.3390/agriculture12050722_

Round 1

Reviewer 1 Report

The migration of labor force between urban and rural areas is a normal social and economic phenomenon. Therefore, the author chooses a very meaningful research topic. A few comments for the author's reference: 

(1) The introduction needs to be rewritten. The present introduction fails to recognize the marginal contribution of the research and the key scientific questions to be addressed. 

(2) The theoretical analysis part needs further strengthening. In fact, there have been many studies on labor migration between urban and rural areas. The core theories include Todaro's migration model and the new migration economics theory. However, the author has not well combined these classical theories with the research design. 

Xu, D., Deng, X., Guo, S., & Liu, S. (2019). Labor migration and farmland abandonment in rural China: Empirical results and policy implications. Journal of Environmental Management, 232, 738-750.

(3) The conclusion and discussion section need to be further condensed. For example, what are the similarities and differences between this study and similar studies? What are the reasons for the differences? In addition to the policy implications for the region being studied, what implications could this study have for other regions? 

Author Response

Dear Reviewer,

We are very grateful that you have made such constructive and substantial suggestions to improve our paper. We have addressed all recommendations to strengthen and clarify our approach in the revised version of the article.  Please see the attached table. 

Reviewer 2 Report

The authors investigate the factors that determine farmers' choices to migrate from rural to urban areas in two regions in Bangladesh. They identify six dimensions which are critical —individual, household, economic, attitudinal, spatial, and climate-related— and provide a number of recommendations to prevent its excessive increase.

The article is clearly written and straightforward. Methods are appropriate and clearly described. Results and conclusions are valid and useful, and they answer the aims of the study. I just have a couple of broader concerns. In the first place, the introduction should include the justification for conducting the study, the paper’s objective and its contribution. It is now included in Section 2. Second, the section on theoretical background, though interesting, is somewhat too dense. I would suggest somehow structuring it better.

I include below some minor comments which need to be addressed:

  • Lines 13-15: The sentence “It is frequently discussed and cited about the factors, patterns, causes and consequences, socio-economic and environmental impact of migration from the general perspective” could be somewhat more clearly written.
  • Lines 18-20: The sentence “But why farmers migrate to the urban area/or are there different factors responsible for farmers rural-urban migration is a prior concern to academia and key actors, where the country’s economy mainly depends on agriculture and farming” could also be more clearly written.
  • Line 25: The authors state here that 50 variables were selected, whereas later they say that only 30 or 20. I would suggest checking this and correcting it where necessary.
  • Line 44: I would suggest specifying where “About 740 million people are internal migrants”. I mean, is it in the world?
  • Line 193-194: The sentence “However, these variables were inconsistent because of the locational variation in the case studies” is kind of confusing. Is it really necessary to include it?
  • Line 214: The authors state that “a list of 30 variables” were selected. The same applies as above. I would suggest checking this and correcting it where necessary.
  • Line 223: The needed citation is still missing.
  • Line 257: I see more than 20 variables in Table 2.
  • Lines 260-261: The sentence “Q-methodology for also elicited subjective viewpoints and identifying shared patterns among the local farmers” is not complete.
  • Lines 271-285: I think the variable “anticipated jobs in urban area” has been written twice under two different headings. The variables “education” and “low price of agricultural products” have not been included in Table 2.
  • Lines 301-303: These two sentences seem to be somewhat contradictory.
  • Lines 325-327: The authors state that “Higher household income levels may ensure…” Is this conducive to R-U migration? I would suggest specifying this.
  • Line 376: The sentence “moreover, land use policy” is not complete.
  • Line 388: According to Table 3, this dimension consisted of four variables, not three as stated here. The communality and the factor loading of the fourth variable are also missing in lines 391-392.
  • Lines 398-402: Why have these specific factors been listed here? They are not necessarily the most important ones according to Table 2.
  • Lines 405-406: Even though the authors state that “There should thus be two different kinds of investment in the rural areas”, they seem to then explain three different kinds of interventions.
  • Lines 412-413: I would suggest checking the sentence, it leads to confusion.
  • Not all references are presented in alphabetical order.

Author Response

Dear Reviewer,

We are very grateful that you have made such constructive and substantial suggestions to improve our paper. We have addressed all recommendations to strengthen and clarify our approach in the revised version of the article. Please see the attached file.

Round 2

Reviewer 1 Report

i have no other comments.